# Meanings and Practices in Intercultural Health for International Migrants

**DOI:** 10.3390/ijerph192013670

**Published:** 2022-10-21

**Authors:** Consuelo Cruz-Riveros, Alfonso Urzúa, Gustavo Macaya-Aguirre, Báltica Cabieses

**Affiliations:** 1Escuela de Psicología, Universidad Católica del Norte, Antofagasta 1270709, Chile; 2Escuela de Enfermería, Facultad de Salud, Universidad Santo Tomás, Concepción 3460000, Chile; 3Facultad de Ciencias Sociales, Universidad Alberto Hurtado, Santiago 8320000, Chile; 4Facultad de Medicina Clínica Alemana, Universidad del Desarrollo, Santiago 7610658, Chile

**Keywords:** interculturality, health care, health personnel, health system

## Abstract

In this article, meanings and practices in intercultural health for international migrants in health establishments are described from the perspective of health personnel in the city of Antofagasta in northern Chile. Methodology: The methodology was qualitative with a phenomenological descriptive design, through which discourses from health personnel in the public primary and secondary care system were explored (*n* = 23). Next, meanings and practices in intercultural health for international migrants in health establishments are described from the perspective of health personnel in the northern Chilean city of Antofagasta. Results: The participants presented trees of thematic categories. There were three thematic categories overall: (1) The meaning of interculturality included features of understanding of the concept, with respect for culture being the transversal axis in all discourses. (2) Practices in health care, where voluntariness, references, and the adequacy or non-technicality of the language are fundamental axes. (3) Training in the intercultural approach, where there is often self-knowledge and lack of supply in the health system. Conclusions: The findings show essential elements to consider in the care of international migrants, including the training and awareness of staff about the intercultural approach through strategies following the local reality in which each health establishment exists.

## 1. Introduction

As a mobility phenomenon, migration has accompanied people since ancient times [1]. The World Health Organization (WHO) defines it as the movement of population from one territory to another, thereby implying the crossing of administrative borders to another state [2]. As of 2020, 280 million people resided in a country other than the country of birth [3]. In Chile, the international migrant population is close to 8.6% of the total, with the three regions with the most significant amount of the migrant population being the Metropolitan with 63.1%, Antofagasta with 7.1%, and Valparaíso with 6.4% [3,4,5].

Although the factors triggering migration are diverse, including political, economic, social, environmental, educational, and health factors [1], poor living conditions in the countries of origin, such as poverty, unemployment, and violence, are relevant to Latin America. Social and natural disasters [6], regardless of the cause, such as migration constantly have effects on people, including both physical and mental health [2,5].

Among the factors that positively or negatively impact the health status of the international migrant population, the WHO has indicated the presence of good health status as a protective factor in much of this population [6,7,8,9,10]. Despite this, there are other risk factors present during the displacement stage and/or during the stay in the receiving countries. These include living in conditions of poverty, overcrowding, poor working conditions, and difficulty in accessing and using social and health services [6,7,8,9,10]. The repercussions produced in the living conditions they acquire in the receiving country have directly contributed to the increase in health problems, such as chronic non-communicable diseases and mental health (e.g., depression and anxiety) [1,6].

Approaching health from the migratory perspective can be carried out by analyzing the endogenous or exogenous factors affecting health status [2,4,6,8]. Among the exogenous factors that have a proven impact on the health of migrants is the health system, with health practices and the implementation of theoretical aspects inserted in care programs causing satisfaction or dissatisfaction with the service provided [2,4]. One factor that has presented critical elements associated with the health system is the interpersonal relationships which occur in the health establishment during health care. This may reflect the poor implementation of health policies [2,4,6,8].

In response to the various problems experienced by health systems in the different recipient states, the WHO has generated guidelines and strategies whose central elements are to consider health as a right, and to respect users’ values and beliefs [1,8]. This implies considering the health dynamics between health teams and migrants’ dimensions of the right to health, such as acceptability, which is understood as “respect for medical ethics and culturally appropriate, taking into account age and gender” [11]. According to the different components of the health model and the approaches that allow guiding practices, considering people as social beings with rights and duties requires practices that consider professional, ethical aspects based on mutual respect [1].

One strategy proposed by the WHO is a practice based on the intercultural approach, which requires understanding essential theoretical aspects for respecting the right to health by service providers [12]. For this, interculturality can be understood as a form of coexistence between people, groups, and institutions with diverse characteristics and cultural positions, characterized by coexisting and relating in an open, horizontal, inclusive, respectful, and synergistic manner in a shared context [12], which support health practices that consider the intercultural approach based on four basic principles: recognition of diversity, respect for differences, equitable relationships, and mutual enrichment [4,13,14]. The central axis is the recognition of migrants as subjects of law, holders of dignity, and as potential sources of wealth due to their diversity [13]. Although there are currently theoretical and legislative frameworks for intercultural practices, they would present problems in the implementation that health teams must carry out in topics related to acceptance of the other, respect, inclusion, equity, reciprocity of benefit (personal and users), and solidarity for an efficient implementation [13,14].

The strategies that each national health system should include are related to the direct care of the health teams (service providers) who perform direct care [13,14]. Some factors recognized in previous studies as barriers in individual practices are voluntariness and willingness to integrate adaptations during health care [4,5,13,14,15,16,17]. The cause could be associated with intervening factors, such as little knowledge about interculturality and individual and collective beliefs about the international migrant population [4,5,13,14,15,16,17]. These factors result from poor implementation of the intercultural approach in health practices and perception of dissatisfaction with the care received by international migrants [4,5,13,14,15,16,17].

According to the above, although it is necessary to acquire knowledge about interculturality, it is essential to have transversal strategies that give rise to implementation of deliberate and sustained practices in different health systems [18]. It is precisely the health personnel where the meanings and conceptions of interculturality are embodied in practices, which can be diverse and even trigger difficulties in meeting international migrants’ health needs [18,19,20]. Among the reported practices are discriminatory acts, standardization of care, ignorance or erroneous knowledge of protocols, denial of access, and not respecting the whole exercise of the right to health for not presenting documents [18,19]. According to the above, although it is necessary to acquire knowledge about interculturality, it is essential to have transversal strategies to give rise to implementing deliberate and sustained practices in different health systems [18].

Indeed, health responses delivered to international migrants that do not consider the intercultural approach for its implementation can trigger inadequate management of health needs [5]. Consequently, there would be a deterioration in the health status of the individual and/or their relatives [5]. In response to the background presented above, the following study aims to describe meanings and practices in intercultural health for international migrants in health facilities, from the perspective of health personnel in the city of Antofagasta, in northern Chile. The guiding questions are: How do health personnel understand care with an intercultural approach for international migrants in primary health care? What are the practices in health care with an intercultural approach for international migrants in the Chilean health system?

## 2. Materials and Methods

### 2.1. Design

A phenomenological-descriptive qualitative design was used to achieve the purpose of the study, to explore the discourses and descriptions of meanings and practices of health personnel. The use of the interpretive approach presented by qualitative research allows us to make sense of and interpret meanings and practices from each individual reality; for this study, we looked at health personnel [21].

### 2.2. Participants

Convenience sampling was used, associated with the permits granted by the health authorities for researchers’ entry, for which a list of people available at the time and place was delivered who participated voluntarily [22]. Snowball sampling was used to reach a more significant number of participants as well. Its main characteristic is the participation of personnel invited directly by the seed participant, generating greater confidence and comfort when talking about their experience as a participant [22]. The selection required both techniques due to officials’ low participation. Data saturation criteria considered two elements: (i) new data provided and (ii) researchers’ experience.

A total of 23 people participated (13 in individual interviews and 10 in focus groups), all from the public primary and secondary health care system. Interviews ran from April to August 2021 (Table 1). The inclusion criteria were (i) being over 18 years of age and (ii) having provided health services to migrants. Among the interviewees were two authorities, one from the primary level and one authority from the health service (Table 1).

### 2.3. Data Collection

This was carried out through in-depth interviews (13 participants) and three focus groups (10 participants). Both techniques were chosen to fulfill the purpose of the investigation. The in-depth interview as a technique has the benefit of being face-to-face or one-on-one between researcher and participant, requiring commitment, willingness to participate, and decreased desirability in responses that would act as a bias, which increases with the focus group [23]. Another benefit of the in-depth interview is fairness in the time given to each participant compared to the focus group technique [23,24]. The advantage of the focus groups is the richness of meanings it can provide; it also facilitates openness and spontaneity of expression [23,24].

The data was collected in participants’ workplaces, ensuring sanitary safeguards, a calm environment, privacy, anonymity, and confidentiality. The interview guide used questions about the approach to health practices, the meaning of interculturality, and training on the subject. Some of the questions used in the focus groups are: Has anyone had training problems in health and interculturality? What is intercultural health? Are concrete actions implemented or in conjunction with the work team to serve migrants? Some of the questions used for in-depth interviews are: What is interculturality? Have you had any training in health and intercultural issues? Interviews lasted between 40 and 60 min and were digitally recorded, converted to audio files on a laptop, transcribed verbatim, and then analyzed.

### 2.4. Rigor

Rigor is guaranteed via triangulating information, given by using two techniques (in-depth interview and focus groups) that deliver similar results, which allows for corroborating the findings [24,25,26]. The techniques used, such as in-depth interviews and focus groups, also seek to broaden the perspectives presented by the participants, and the questions used are semi-structured [24,25,26]. Different perspectives presented by health personnel with different hierarchies and activities within the establishment are included as well. The audit and reflection were carried out by another researcher who followed the process step by step and the decisions used, reaching similar conclusions [24,25,26]. Expert researchers followed the generation of the question script.

### 2.5. Ethical Considerations

The study was approved by the ethics committee at Universidad del Desarrollo. It was governed by the principles of voluntariness, confidentiality, and participant anonymity, which was reflected through the signing of informed consent.

To obtain the participants in the primary care centers, the objective of this study was presented to the directors, and information was provided on data generation methods and confidentiality agreements. All participants provided signed informed consent. Before signing the informed consent, the participants were given information about the purpose of the research and their rights. Informed consent was then signed, with the date and time of the interview. Codes were used to ensure anonymity and confidentiality as well.

### 2.6. Data Analysis

After the interviews were transcribed into a Word document, thematic analysis was carried out, as done in previous similar studies [27]. This thematic analysis consists of repeated readings of the transcripts focused on identifying main categories and related themes. The process described above was carried out with interviews and focus groups, to be later integrated into a matrix. The broad categories were coded from the matrix through the construction of analysis categories assigned by the researchers considered at the time the interview script was drafted; despite the above, the review expects to find emerging codes and assign them one or more related codes with their meaning and intent. Subsequently, the process called “decantation” was carried out following the open coding, then the axial coding, and finishing later with the selective coding. The analysis was carried out and supervised by all the authors, and coding progress required the approval and considerations provided by the team.

## 3. Results

The information provided by the participants focused on three categories; (1) One category is the meaning of interculturality, which included the subcategory understanding of the concept, the main characteristic being respect for culture, which is the transversal axis in all discourses. (2) In practices in health care, two subcategories emerge. The first is health personnel, characterized by practices based on voluntariness, including in-care actions such as language adaptation or not using technicalities, which are fundamental axes, and the second subcategory, the health system, characterized as where the inclusion of intercultural referents as support has facilitated the understanding and care focused on the needs, although there is little or no adaptation of the protocols to the existing resources in the establishment. (3) Training in intercultural approach and self-knowledge for health personnel is frequently required by the health system.

1.
*Meaning of interculturality*


The exploration of the speeches presented are with respect for the other as a central axis, characterized by rescuing the value of the reality experienced by international migrants during their lives. This necessarily includes respect for the culture and with it the knowledge they possess, their training, their beliefs, and understanding of the influence that the environment has presented. Therefore, interculturality in health practices should consider language and cultural adaptation to respond to the needs of people in a timely and relevant manner.

*“… Intercultural health is mostly respecting the worldview of the other being that we can’t, uh, root out what one already knows and impose another, uh, do something more complementary, both respecting decisions”*.(EHP1)

*“… Interculturality is all we can rescue from all areas, places, people, in general from all formations… Most of all, they are of their behaviors, of their formation, what they carry internally as part of life, is what they believe of life or are within their life, their reality”*.(EHP2)

*“… The work that has to be done with it will always be focused, and in its entirety, understanding and respecting its origins, its culture, its beliefs to be able to do the work”*.(EAPS1)

*“… We have to be aware that that person is culturally, his environment influences his health and finally we have to be aware of that, and we adapt our benefits and understand the state of that patient, being able to be biological, social or psychological, from that point of view”*.(EAPS2)

2.
*Practices in health care*


The stories focused on the voluntary implementation of practices at the local level. Although elements delivered as public policies were manifest in the speeches, the implementation in the health centers embodied in documents appropriate to the local reality did not materialize. The difficulty of the above triggers individual practices that were often inherited from other members. One positive aspect achieved in some centers was the inclusion of references that helped orient the team regarding some problems. Participants also highlighted the lack of scientific support for the generation of official documents on the practices to be carried out by some professional bodies, guiding the services to established standards focused on biological elements.


*“… The role of facilitator is a link both to the indigenous peoples in my case, and to the infrastructure or, or more traditional health, is a link.”*
(EAP1)

*“… There would have to be a more complex sustenance and more scientific information to be able to make protocols…. as I say, physiology, pathophysiology and physiology, there must be greater studies of sustainability”*.(EHP2)

*“… In the unit we work with the user advisory councils and there are, uh, people of different nationalities and of different cultures, in fact, they, when we do business meetings”*.(EHGP4)

*“… I think we’re each working with what you think in your head and believe, like that concept… The attention in the SOME yes, they are practices so individual that it is as if there was something blunt that we will all work alike, it is more that each with its conditions are put in the place of the other, but the SOME is so diverse, but so diverse that they are so inadequate sometimes in that space”*.(EAPS1)

*“… The referent, she is the one who supports us in every sense on the part of migrants, so any consultation or something I see with her, plus the administrative part and there it is seen… You have to adapt to their language during visits”*.(EAPS2)


*“… it doesn’t exist at all… By the way you said to me you ask me, listen, what if I try this, and it’s like, if you think that it does you good, well, it’s good.”*
(EAPS7)


*“… I have had to adapt to each of my patients in a different way, even, not only talking about a cultural issue, at the level of I don’t know, of nationality, but also education, I cannot explain with the same technicalities.”*
(EAPS10)


*“… There is nothing, there is no written protocol of… In case you have to do this… no, not really.”*
(EGHP1)

3.
*Training in intercultural approach*


The speeches reveal a lack of formal training, self-knowledge, and the support received by intercultural referents. It also establishes the need to integrate the entire health team and requires a training plan from the authorities.


*“… At least in my case, when I became a reference point for migrants, I did not have much knowledge of the truth on the migration issue, very little, and… of course the colleague.”*
(EGHP4)


*“… Like, learning as cases have come along, to be honest.”*
(EGHP1)

*“… But this does have to address the whole health team and this has to be done annually, just as you are required to have the first aid course to train and accredit. We should have this outlook and this training internalized in our establishment, given the situation in our region”*.(EAPS4)

## 4. Discussion

The study addressed meanings and practices in intercultural health for international migrants in health establishments, from the perspective of health personnel in the northern Chilean city of Antofagasta. Primary and secondary care levels were considered for this study.

The first guiding question of the study was: How do health personnel understand care with an intercultural approach for international migrants in the Chilean health system? The participants pointed out the meaning of interculturality, understanding the concept as the exercise of respect for migrants during the health care process, considering their life history, beliefs, and personal characteristics (also emphasizing the educational level). According to international organizations such as the Pan American Health Organization (PAHO) and the WHO, it is essential to consider the intercultural approach in the health system when implementing regulations and programs that are part of health establishments’ service portfolio [14]. Equitable and respectful interactions of social, cultural, ethnic, gender, and life cycle contexts can be obtained through these actions [8,12,14]. On the other hand, the normative aspects currently established worldwide have received criticism regarding the contradiction between the declaration of the right to health as universal and equal and interculturality, which considers the dimension of acceptance for satisfaction as the central axis of health needs in the international migrant population [28,29].

The second guiding question was: What are the practices in health care with an intercultural approach for international migrants in the Chilean health system? According to previous research, language and cultural adaptation are recognized as potential barriers the health system presents for international migrants’ care [30,31,32,33]. Health personnel consider including aspects such as language and cultural adaptation in their practices, one of the most recurrent being the adaptation of language without technical jargon. The evidence also highlights the result of culturally competent interventions (considering among them the language adaptation of institutional documents and infrastructure, such as language adaptation for signs) to achieve greater adherence to treatments, continuity of care, and a better perception of service quality [28]. Authors such as Urrutia et al. highlight that health workers carrying out intercultural meetings need to develop skills that can allow for an action-oriented understanding of the reality of the other, as a fundamental element for health practices based on interculturality [34]. Indeed, the multiple practices that allow the exercise of the right to health based on understandable information, the orientation of administrative processes, and acceptance of the other all allow for maintaining or improving the health status of people and those close to them.

When considering the discourse from the participants about training in the intercultural approach, a small amount of formal training was reported to be available, which results in the development of self-knowledge, thus triggering practices that they name as trial and error. Another way of acquiring knowledge or information from other health team members is the transfer of information and practices carried out by others. Among the most relevant is the support received by the intercultural referent. According to previous studies, as reported by Sepúlveda et al., among the currently highlighted intervention strategies, the role of the facilitator has been relevant for supporting both health personnel and international migrants [29].

In our findings, together with knowledge about interculturality for health care, we found two crucial elements for implementing practices in establishments. Among them was the will of the health worker and the transfer of knowledge between the team, sometimes given by trial and error. We were generating practices that can vary within the enclosure, with those oriented in the biomedical model remaining a common axis.

Regarding the meanings reached by health personnel, health practices can be permeated by not having expert training and guidance. Without the necessary training, our own culture that presents ideologies, values, beliefs, prejudices, and knowledge is established as an element of comparison, where we ultimately obtain an interpretation influenced by the other and its context, wherein the concepts of health and disease are established from the perception of reality of the health worker [35].

In summary, the meanings and practices in health care for international migrants present factors that can act as barriers or facilitators of the health system. This is relevant to the intersectoral work that involves both the authorities and health personnel for developing actions based on local realities. Despite these, one of the strategies that can improve staff implementation of policies and regulations is training in interculturality.

The limitations of the study are as follows: The scope is restricted, mainly given the geographical limitation of the study. The countries of origin of the vast majority of international migrants within northern Chile are mainly from South America. In future research, it is pertinent to expand the sample to the different national regions and include the different levels of health personnel (auxiliary, administrative, and professional). In addition to the above, the design limitations do not allow the results to be generalized. Regarding researcher bias, this type of research is about minimizing, hence the declaration of default procedures and data collection methods.

## 5. Conclusions

This study manages to establish that the health system needs to achieve health practices based on the intercultural approach. In conclusion, the findings show fundamental elements for health practices with an intercultural approach, among which are staff training and awareness. Another relevant aspect to consider is the implementation of strategies both at the national and local levels by the local reality where each health facility exists.

## Figures and Tables

**Table 1 ijerph-19-13670-t001:** Participants’ attributes.

Code	Role	Sex	Age
EGHP1	Matron	Female	* Greater than 25
EGHP2	Matron	Female	33
EGHP3	Social worker	Female	29
EGHP4	Social worker	Female	28
EHP1	Cultural facilitator	Male	23
EHP2	Matron	Female	28
EAPS1	Psychologist	Female	36
EAPS2	Nurse	Female	* Greater than 25
EAPS3	Administrative	Female	56
EAPS4	Matron	Female	31
EAPS5	Nurse	Female	40
EAPS6	Psychologist	Female	36
EAPS7	Kinesiologist	Male	39
EAPS8	Nurse	Female	39
EAPS9	Social worker	Female	31
EAPS10	Matron	Female	* Greater than 25
EAAPS1	Technical reference	Male	51
EGAPS1	Physician	Female	28
EGAPS2	Nursing Technician	Female	33
EGAPS3	Matron	Male	25
EGAPS4	Nurse	Female	36
EGAPS5	Psychologist	Female	28
EGAPS6	Dentist	Male	31

* No age specified.

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
