# Peer review of "Meanings and Practices in Intercultural Health for International Migrants"

_ijerph, 2022, doi:10.3390/ijerph192013670_

Round 1

Reviewer 1 Report

This study described the meanings and practices in intercultural health used by health personnel in the care of international migrants through questionnaire interview and qualitative analysis. I recommend a major revision.

1. The English writing quality in some section is poor, and some sentences are hard to understand (e.g., 96-100). Please correct the mistakes (e.g., Abstract; Title 2.1), and polish the language and sentences. 

2. The factors that act as a barrier in the health system for the implementation of the intercultural approach in health practices could be categorized into three groups including health system government factors, resource availability, and service providers. Why conducting the investigation solely by health personnel (i.e., service provider).

3. The research gaps are not clarified. What is contribution of the research objectives compared with previous studies.

4. What is the qualitative coding framework relevant with the interview questions?

Author Response

Please write down "Please see the attachment".

Reviewer 2 Report

Dear Authors

The article is undoubtedly interesting and relevant in terms of public health, however, during the review I had some doubts that are not fully resolved in the document. Below, I will present my comments according to each section.

Introduction

In general, it is considered that the introductory section contains enough information to problematize and give context to the object of study, it is even possible to see that there are elements that could serve as a contrast for the discussion of the results obtained, for example, the concept of intercultural health proposed by the WHO in contrast to the symbolic constructions of the study participants.

I would just like to point out that on lines 43-47 and 14 and 15 the font has a smaller size.

Materials and methods

It is understood that the study has a qualitative methodology with a phenomenological approach, however, the reasons why the researchers divided the data collection techniques are not clear. On the one hand, it is argued that the in-depth interview has advantages over focus groups, however, the advantages of using a focus group over an interview are not made explicit. In this sense, it is suggested that the researchers explain the reasons why these two data collection techniques were used.

Likewise, it is necessary to explain the advantages/reasons of using a “snowball” type of sampling in contrast to another type of non-probabilistic sampling, for example, the convenience that would have facilitated the identification of different key actors for the study.

In the methodology the criteria that were considered to reach the theoretical saturation are not established, it is widely suggested that the researchers describe the criteria under which the saturation point was determined. Given that the sample is very diverse, a wide thematic and ideological diversity would be expected, however, in the results it is not possible to visualize this, therefore I think it is important to assess the method and its results in light of the criteria that were considered to establish a limit in the search for additional information in the informants.

It would be advisable for the researchers to clarify the reasons why they state that it was difficult to obtain a sample of health workers (line 147). Likewise, it is advisable to indicate the reasons why it was not possible to retrieve the sociodemographic information of the complete sample. Strictly speaking, the sample size is small and should not represent an additional difficulty in retrieving complete information.

In relation to triangulation, it is not completely clear how it was carried out, for example, was it triangulated with different researchers with different methods? With different analytical categories? Or research approaches? Researchers are kindly requested to explain the triangulation mechanisms as well as the elements that were methodologically modified and that eventually led to similar conclusions.

In the data analysis it is clear how the information derived from the interviews was processed, however it is not explained how the information from the focus groups was analyzed, please suggest that it be explained.

Finally, it is strongly suggested that researchers include the semi-structured interview protocol that was used as an annex.

Results

The citations selected by the researchers may not be self-explanatory or probably require some translation adjustments, since convergence between the explanation and what the citations point to is not observed. I strongly suggest revising the style of the language so that the exposition is clear and it is possible to see the links between the citations and the interpretation provided by the researchers.

Regarding the style, it can be seen that from line 244 to 286 the font changes and is larger than the rest of the text.

In line 240 a reference is made to a person, but it is not clear who.

Once again, it is suggested that the translation or adaptation of the language from Spanish to English be reviewed, since the productions are not clear and it is not possible to analyze the content of the quotation, for example in lines 244, 252, 253, and thus in the following quotes.

In line 274 the information is clearer and it is possible to visualize the link between the interpretation and the quotes, however, I think it would be convenient to place more quotes, more evidence, which demonstrates the theoretical saturation and that the informants are indeed talking about the same topic.

Discussion

I believe that the results must be put in context with respect to the recommendations issued by international organizations regarding the findings obtained and what is indicated from the theory.

Author Response

(The authors gave the same response as above.)

Reviewer 3 Report

Formal aspects:

The type of letter is not correct nor adapted to the standards of the journal. There are different fonts in the text.

What were the inclusion and exclusion criteria of the individuals?

What checklist was used to verify the quality of qualitative research?

What data analysis program was used?

Author Response

(The authors gave the same response as above.)

Round 2

Reviewer 1 Report

I recommend accept.

Reviewer 2 Report

Dear Authors 

I have reviewed the document and noticed that most changes were made succesfully. Thanks for consider those observations to improve your manuscript quality. 

I noticed that the interviews are in spanish, could it be possible to translate them? I would be great that other researchers that do not speak spanish could read them. 

Best Regards

Reviewer 3 Report

Thanks for implementing the changes and recommendations and congratulations for the work.